# Anti-Biofilm Activity of Assamsaponin A, Theasaponin E1, and Theasaponin E2 against *Candida albicans*

**DOI:** 10.3390/ijms25073599

**Published:** 2024-03-22

**Authors:** Yuhong Chen, Ying Gao, Yifan Li, Junfeng Yin

**Affiliations:** 1Key Laboratory of Tea Biology and Resources Utilization, Tea Research Institute, Chinese Academy of Agricultural Sciences, Ministry of Agriculture, 9 South Meiling Road, Hangzhou 310008, China; chenyuhong@tricaas.com (Y.C.); liyifan@tricaas.com (Y.L.); 2Tea Research Institute, Hunan Academy of Agricultural Sciences, Changsha 410125, China

**Keywords:** *Camellia sinensis* seeds, assamsaponin A, theasaponin E1, theasaponin E2, anti-biofilm activity, virulence factors

## Abstract

Biofilm formation plays a crucial role in the pathogenesis of *Candida albicans* and is significantly associated with resistance to antifungal agents. Tea seed saponins, a class of non-ionic triterpenes, have been proven to have fungicidal effects on planktonic *C. albicans*. However, their anti-biofilm activity and mechanism of action against *C. albicans* remain unclear. In this study, the effects of three *Camellia sinensis* seed saponin monomers, namely, theasaponin E1 (TE1), theasaponin E2 (TE2), and assamsaponin A (ASA), on the metabolism, biofilm development, and expression of the virulence genes of *C. albicans* were evaluated. The results of the XTT reduction assay and crystal violet (CV) staining assay demonstrated that tea seed saponin monomers concentration-dependently suppressed the adhesion and biofilm formation of *C. albicans* and were able to eradicate mature biofilms. The compounds were in the following order in terms of their inhibitory effects: ASA > TE1 > TE2. The mechanisms were associated with reductions in multiple crucial virulence factors, including cell surface hydrophobicity (CSH), adhesion ability, hyphal morphology conversion, and phospholipase activity. It was further demonstrated through qRT-PCR analysis that the anti-biofilm activity of ASA and TE1 against *C. albicans* was attributed to the inhibition of *RAS1* activation, which consequently suppressed the cAMP–PKA and MAPK signaling pathways. Conversely, TE2 appeared to regulate the morphological turnover and hyphal growth of *C. albicans* via a pathway that was independent of *RAS1*. These findings suggest that tea seed saponin monomers are promising innovative agents against *C. albicans*.

## 1. Introduction

*Candida albicans*, a prevalent pathogen responsible for hospital-acquired infections, presents a significant risk to patients’ lives and well-being, with invasive candidiasis resulting in mortality rates ranging from 40% to 60% [1]. *C. albicans* is classified as a diphasic fungus, and its capacity to transition from a commensal to a pathogenic state is attributed to a diverse array of virulence factors. These factors include signaling molecules that mediate adhesion and invasion into host cells, the secretion of hydrolases, the transition from yeast to hyphae, the formation of biofilms, and various other attributes [2]. Notably, the capacity for morphological transformation and biofilm formation is a key property of *C. albicans* [3]. The majority of *C. albicans*-related infectious diseases are closely associated with the presence of biofilms within the host or on non-biological surfaces [2]. Biofilms increase the likelihood of systemic infection. The biofilm architecture of *C. albicans* is recognized as a contributing factor to high patient mortality [4]. Biofilms are intricate fungal communities, comprising a compact three-dimensional framework of *C. albicans* in three distinct morphological states (yeast, pseudohyphae, and hyphae) enclosed within an extracellular matrix [5]. The formation of biofilms is divided into four distinct stages. Firstly, adhesion occurs when yeast cells attach themselves to either cellular or non-biological surfaces with the help of adhesins and other substances. Following adhesion, yeast cells undergo proliferation and generate reproductive tubes, which is a stage known as initiation. As the biofilm matures, pseudohyphae and hyphal development take place, accompanied by the accumulation of extracellular matrix. Finally, at the diffusion stage, yeast cells disperse from the biofilm and colonize new locations [6,7]. Due to their intricate architecture, biofilms are commonly regarded as a survival mechanism employed by microorganisms. Currently, there are five main types of antifungal drugs against *C. albicans* biofilms based on their action sites: azoles, polyenes, allylamines, candins, and flucytosine [8]. Numerous studies have consistently demonstrated the inherent resistance of cells within biofilms to these conventional clinical therapeutic drugs as well as their immune evasion ability [9,10,11,12]. In addition, the emergence of more drug-resistant strains requires alternative therapeutic agents to combat biofilm-mediated infections and overcome the limitations of current antifungal therapies. Notably, *C. albicans* cells in biofilms exhibit a resistance level to antifungal drugs that surpasses that of their planktonic counterparts by a factor of 1000 [11]. Mature biofilms exhibit formidable resistance against medical interventions, including disinfectants and physical washing [13]. Consequently, inhibition of biofilm formation emerges as a crucial strategy for mitigating fungal infections and resistance.

*Camellia sinensis* seed saponins are a class of non-ionic surfactants consisting of pentacyclic triterpenes, with a content of over 10% in tea seeds, and possessing various excellent biological activities. Our previous research has demonstrated the fungicide effects of tea seed saponins on planktonic fluconazole-resistant *C. albicans* [14]. These saponins inhibit ergosterol biosynthesis, leading to the disruption of cell membrane morphology and function. Additionally, they regulate crucial energy metabolism processes, such as intracellular glycolysis [14]. However, there is still a lack of research on the effects of tea seed saponins on the pathogenic biofilm state of *C. albicans* and the underlying mechanisms.

To investigate the impact of tea seed saponins on the virulence factors of *C. albicans*, the inhibitory activities of three tea seed saponin monomers—theasaponin E1 (TE1), theasaponin E2 (TE2), and assamsaponin A (ASA) (Figure 1)—on the adhesion, morphological transformation, and biofilm formation ability of *C. albicans* ATCC 10231 were assessed. Additionally, the inhibitory mechanisms of tea seed saponins on *C. albicans* biofilm were elucidated by analyzing the expressions of *C. albicans*-specific genes. The results help answer the question of whether tea seed saponins possess the potential to attenuate the pathogenicity of *C. albicans*.

## 2. Results

### 2.1. The Biofilm Growth Curve of Biofilm Formation of C. albicans ATCC 10231

The process of *C. albicans* biofilm formation is usually categorized into three distinct stages: the early stage (0–11 h), the middle stage (12–30 h), and the mature stage (31–72 h) [7,15]. The biofilm formation capacity of *C. albicans* is influenced by various factors, including strain types, growth conditions, and attachment materials [16,17,18]. Therefore, to evaluate the biofilm formation potential of *C. albicans* strain ATCC 10231, an investigation was undertaken to ascertain its ability to develop biofilms on plastic polystyrene surfaces at different incubation time periods. The findings, illustrated in Figure 2, demonstrate that *C. albicans* commenced adhesion after 1.5 h, followed by a gradual increase in biofilm growth until reaching a plateau at the incubation time of 24 h. Notably, there was no statistically significant difference observed between the biofilm growth at 48 and 72 h (*p* > 0.05), indicating that this strain had the ability to form fully developed biofilms within 24 h, which is aligned with the static biofilm model established by Ramage [19]. Thus, the incubation time of 24 h, at which the mature biofilm was initially formed, was selected for subsequent investigations.

### 2.2. ASA, TE1, and TE2 Inhibit the Adhesion of C. albicans

Adhesion is the first step in the biofilm formation of *C. albicans*. ASA, TE1, and TE2 significantly inhibited the adhesion of *C. albicans* to the surface of polystyrene in a concentration-dependent manner (Figure 3). The 50% inhibitory concentrations (IC_50_s) of ASA, TE1, and TE2 regarding the adhesion of *C. albicans* were 32.87, 33.64, and 44.70 µM, respectively, indicating that the compounds ranked as follows in terms of inhibitory activity: ASA > TE1 > TE2. Our previous research established that TE2 did not possess fungicidal properties against planktonic *C. albicans* ATCC 10231 at concentrations ranging from 0 to 800 µM [14]. In this study, it was observed that TE2 led to a reduction in adhesion ability to 68.24 ± 1.03% at a concentration of 100 µM. However, the adhesion inhibition rate did not increase as the concentration of TE2 increased, suggesting that TE2 might not inhibit adhesion through a reduction in viable cells. This was further validated via microscopic examination (Figure 3C). When the concentration exceeded 100 µM, the cell numbers under ASA and TE1 decreased, whereas TE2 did not result in this trend. Moreover, Figure 3C shows that 50 µM ASA, 50 µM TE1, and 100 µM TE2 effectively inhibited the yeast state transition into the hyphal morphology of *C. albicans* within 1.5 h. The above results indicate that tea seed saponin monomers attenuated the adhesion by inhibiting the yeast-to-hyphae transition of *C. albicans*, and different monomers exhibited different efficacies.

### 2.3. ASA, TE1, and TE2 Inhibit the Biofilm Formation and Mature Biofilm of C. albicans

Biofilms are the main cause of candidiasis. The XTT reduction assay and crystal violet (CV) staining assay showed that ASA, TE1, and TE2 inhibited the biofilm formation and destroyed the mature biofilm of *C. albicans* (Figure 4). ASA, TE1, and TE2 all reduced the metabolic activity of biofilm cells in a concentration-dependent manner (Figure 4A). Specifically, the ASA, TE1, and TE2 treatments at 25 µM resulted in decreases of 24.44%, 17.85%, and 5.04% in the biofilm metabolic activity, respectively (*p* < 0.05). Treatments with 100 µM ASA or TE1 almost completely inhibited the biofilm metabolic activity of *C. albicans*. It was previously found that the MICs of ASA and TE1 against the planktonic *C. albicans* of the same strain were both 100 µM [14], suggesting that the killing activity of ASA and TE1 was not weakened by the biofilm. Additionally, TE2 demonstrated a 50% inhibition rate on the metabolic activity of *C. albicans*. Furthermore, the CV staining assay demonstrated a concentration-dependent inhibition of biofilm biomass by tea seed saponin monomers. However, the total biofilm amount ceased to decrease with increasing concentrations of ASA, TE1, and TE2 when their concentrations exceeded 100 µM (Figure 4B).

As shown in Figure 4C,D, the tea seed saponin monomers possessed the ability to disrupt the formation of mature biofilms in a dose-dependent manner. The XTT reduction assay showed that 25 µM ASA, 50 µM TE1, and 100 µM TE2 exhibited significant reductions in the viability of *C. albicans* cells (*p* < 0.05). The findings from the CV staining assay, which assessed the overall quantity of biofilm, were in agreement with the results of the XTT reduction assay.

In addition, the 80% inhibitory concentrations of the biofilm (BIC_80_s) of ASA, TE1, and TE2 were 44.62, 71.96, and 148.58 µM, respectively. The 80% eradicable concentrations of the biofilm (BEC_80_s) of ASA, TE1, and TE2 were 113.63, 234.75, and 420.81 µM. The BEC_80_ of ASA, TE1, and TE2 increased by 255%, 326%, and 283% over BIC_80_, respectively. The above results show that the biofilm inhibition and eradication effects of the three tea saponins followed the order ASA > TE1 > TE2, and the eradication concentration was much higher than the inhibitory concentration, suggesting the resistance of mature biofilm was much higher.

### 2.4. ASA, TE1, and TE2 Reduce Cell Surface Hydrophobicity and Extracellular Phospholipase of C. albicans

Cell surface hydrophobicity (CSH) is regarded as a potential virulence factor [20]. Hydrophobic fungi have the ability to facilitate the attachment of *C. albicans* biofilm to various biological and non-biological surfaces and interfaces during the initial adhesion phase, thereby augmenting virulence in mice [21]. In addition, hydrophobic cells are also believed to play a crucial role in the diffusion stage of mature biofilms, potentially serving as key contributors to biofilm formation [22]. The CSHs of *C. albicans* at the adhesive stage with ASA, TE1, and TE2 treatments at a concentration of 100 µM decreased by 30.14%, 24.41%, and 12.18%, respectively (Figure 5A). Meanwhile, the CSHs of *C. albicans* in the mature biofilm with 100 µM ASA, TE1, and TE2 treatments were reduced by 10.58%, 4.27%, and 2.53%, respectively (Figure 5B); this was less effective than at the adhesion stage. The above results show that tea seed saponin monomers reduced the CSH of *C. albicans* at different stages of biofilm formation, with ASA exhibiting the highest inhibition rate, followed by TE1 and TE2. When comparing this with the results described in Section 2.2 and Section 2.3, it was deduced that tea seed saponin monomers at least partially reduced biofilm formation through a reduction in CSH.

Extracellular phospholipase plays an important role in the invasion and damage of host cells by *C. albicans*, serving as a significant virulence factor during blood-borne infections [22,23]. The results showed that ASA, TE1, and TE2 exhibited concentration-dependent inhibition of extracellular phospholipase production from *C. albicans*, with statistically significant differences observed between the treated and control groups (*p* < 0.05) (Figure 5C,D). Additionally, ASA, TE1, and TE2 also decreased phospholipase viability in mature biofilm cells, although a higher concentration was required compared to that at the adhesion stage.

### 2.5. Effects of ASA, TE1, and TE2 on Genes Associated with Virulence Factors in C. albicans

To elucidate the molecular mechanism underlying the inhibitory effects of ASA, TE1, and TE2 on the biofilm growth of *C. albicans*, the expression levels of genes involved in the signaling pathways of adhesion, biofilm formation, hyphal morphology turnover, secreted hydrolases, and associated transcription factors of *C. albicans* cells were examined (Figure 6).

Figure 6 shows that ASA and TE1 significantly downregulated the expression of adhesion-related genes, such as *ALS1*, *ALS3*, *HWP1*, and *EAP1*, as well as the expression of secreted hydrolase genes, including *SAP5* and *PLB1*, but TE2 could not significantly inhibit the expression of these genes. In terms of genes in the pathways of cellular morphology, hyphal formation, and maintenance, both ASA and TE1 were found to exert significant inhibitory effects on the expression of specific genes, including *UME6*, *EED1*, *HGC1*, *HYR1*, etc. On the other hand, TE2, apart from significantly inhibiting the expression level of the *EED1* gene, did not show significant inhibitory effects on other genes. The activation of hyphal formation can be facilitated by *RAS1* [24], which is subsequently positively regulated by the transcription factor *EFG1* in the cAMP–PKA signaling pathway or by the transcription factor *CPH1* in the MAPK signaling pathway. In addition to the activation of *RAS1*, *C. albicans* can stimulate hyphal growth through alternative signaling pathways. Figure 6 shows that ASA and TE1 exerted an influence on the expression levels of transcription factors, including *RAS1*, *EFG1*, and *CPH1*. TE2 did not exhibit a significant inhibitory effect on *EFG1* and *CPH1* but notably upregulated the expression of *RAS1* in cells. Based on the aforementioned findings, it was inferred that ASA, TE1, and TE2 inhibited the formation of biofilms by impeding adhesion ability and the secretion of proteolytic enzymes. Furthermore, ASA and TE1 exhibited the potential to disrupt hyphal formation and maintenance by inhibiting the cAMP–PKA and MAPK signaling pathways via *RAS1*, whereas TE2 might hinder the hyphal development of *C. albicans* through an *RAS1*-independent pathway.

### 2.6. Effects of ASA, TE1, and TE2 on C. albicans via Addition of cAMP

cAMP is an important component that regulates the development of *C. albicans* hyphae. The addition of cAMP at a concentration of 1 mM effectively restored the mycelial growth of *C. albicans* when treated with 25 μM ASA and 50 μM TE1, as illustrated in Figure 7. In contrast, no notable changes in morphology were observed at a concentration of 100 μM TE2. These findings suggest that the inhibitory impact of ASA and TE1 on *C. albicans* hyphal filamentation is linked to the reduction of intracellular cAMP levels.

## 3. Discussion

*Candida albicans*, a fungus that usually exists in the planktonic yeast form, establishes a symbiotic relationship with humans. However, when the immune system weakens, *C. albicans* undergoes morphological transformation, developing hyphae [25,26,27]. Under the hyphal state, *C. albicans* enhances its capacity to infect the host and evade immune responses, thereby increasing its pathogenicity [27]. Additionally, the ability to form biofilms is crucial to the pathogenicity of *C. albicans* [28]. The intricate structure of biofilms and the shielding impact of the extracellular matrix contribute to the increased resistance of *C. albicans* [5]. Therefore, the primary objective of this study was to examine the impact of tea seed saponin monomers on virulence factors associated with the pathogenicity of *C. albicans*.

The first and pivotal step in the development of invasive infections caused by *C. albicans* is adhesion. *C. albicans* has the ability to adhere to various targets, including human epithelial cells, endothelial cells, mucosal cells, and even non-biological surfaces (like certain medical devices) [29,30]. Research has shown that surfactants are classified into anionic, cationic, zwitterionic, or non-ionic types based on the hydrophilic and hydrophobic binding characteristics of the molecules. Surfactants possess lubricant properties, which enable them to impede biofilm formation by inhibiting microbial adhesion to surfaces and tissues [31,32]. Tawfik et al. indicated that non-ionic surfactants exhibit greater efficacy as antifungal agents compared to both anionic and non-ionic/anionic blend surfactants. This can be attributed to the mechanism of action of non-ionic surfactants, whereby they interact with the binding site through the electronegativity of the oxygen atoms [33]. Tea seed saponins, a natural non-ionic surfactant, have been previously demonstrated to possess favorable antifungal activity. In this study, it was found that tea seed saponin monomers (ASA, TE1, and TE2) at a concentration of 12.5 μΜ significantly inhibited the adhesion of *C. albicans* to non-biological polystyrene surfaces, and the efficacy of inhibition was concentration-dependent. Based on the analysis of the minimum inhibitory concentration (MIC) of ASA and TE1 against *C. albicans* [14], it was found that saponins significantly inhibited the adhesion of *C. albicans* at a concentration of 1/8 MIC. Though TE2 did not possess antifungal activity, it was demonstrated to have the ability to inhibit *C. albicans* adhesion. The adhesion ability of *C. albicans* is directly correlated with its hydrophobicity, which facilitates its attachment to both biological and non-biological surfaces, consequently promoting biofilm formation [34]. In this study, tea seed saponin monomers effectively reduced the CSH, which might be one of the reasons for inhibiting the adhesion of *C. albicans* and reducing pathogenicity.

The adhesion of *C. albicans* is regulated by a variety of protein adhesins present on the cell surface. In comparison to the wild-type strain, the *C. albicans* mutant strain of *ALS1*Δ/*ALS1*Δ exhibited lower pathogenesis and virulence in mice [35,36]. Fu et al. found that strains had an increased ability to adhere to endothelial cells via *ALS1* overexpression [35], while Kamai et al. found that *ALS1*Δ/*ALS1*Δ mutant strains showed significantly reduced adhesion to mouse oral mucosa [36]. This indicates that *ALS1* plays an important role in the adhesion of oral mucosal cells and biological endothelial cells. The protein Als3 plays a significant role in pathogenic mechanisms and has garnered attention as a potential target for vaccines and antibodies against *C. albicans* in recent years [37]. The encoded protein by *ALS3* binds to the E-terminal of oral epithelial cells or the N-terminus of endothelial cells, thereby promoting the adhesion of *C. albicans* to the host [38]. Investigations utilizing a biomimetic human skin cell model showed that the *ALS3*Δ/*ALS3*Δ mutant of *C. albicans* lacked adhesion ability and failed to infect the cells [39]. Hwp1 binds to the glucan on the cell wall surface through covalent bonds, promoting stable and irreversible adhesion between *C. albicans* and the host [40]. Eap1 is also a type of adhesin, and its different domains of protein mediate the adhesion of *C. albicans* to different substrates [41]. In this study, it was found that during the adhesion stage, 100 μΜ ASA and TE1 significantly inhibited the expression levels of adhesin-related genes *ALS1*, *ALS3*, *EAP1*, and *HWP1*, while 100 μΜ TE2 had no significant inhibitory effect on them. Based on the above results, ASA and TE1 weakened the adhesion ability of *C. albicans* to the surface of polystyrene by inhibiting the expression of adhesion genes and CSH. TE2 mainly weakened the adhesion of *C. albicans* by reducing CSH.

The secretion of proteolytic enzymes at the invasion stage of infection is considered a significant virulence factor of *C. albicans*. Common proteolytic enzymes that are beneficial for invasion include secreted aspartate protease (Saps), phospholipase (Pl), and lipase (Lipases, Lip). Phospholipase plays a major role in bloodstream infections, which is potentially linked to its ability to degrade phospholipids in host cell membranes, resulting in host cell damage [42]. Su et al. reported that drug-resistant strains exhibited higher phospholipase activity compared to sensitive strains, suggesting a potential association between phospholipase activity and *C. albicans* resistance [43]. Similarly, Zhou et al. demonstrated a positive correlation between phospholipase activity and adhesion ability [23]. In our study, the main phospholipase secreted by *C. albicans*—phospholipase B—was monitored. In the absence of saponin treatment, *C. albicans* displayed Pz values of 0.43 ± 0.02 during the adhesion stage and 0.56 ± 0.06 in mature biofilms, both indicating extremely high phospholipase activity (Figure 5C,D). Mature biofilm cells displayed lower extracellular phospholipase activity compared to cells at the adhesion stage. The reason might be that phospholipase had a greater damaging effect on the host at the early stage of *C. albicans* infection, and its main function was to promote the formation of hyphae. However, its activity at the late stage of hyphal maturation is unclear and still needs further research [44]. Additionally, the extremely low PZ value suggests that this strain is drug-resistant. Previous literature reports have already established *C. albicans* ATCC 10231 as a fluconazole-resistant strain (https://www.atcc.org/products/10231, accessed on 14 December 2023), a finding that was corroborated by our previous study [14]. The results obtained from the assessment of phospholipase activity in this study further support the resistance of this strain. Following treatment with tea seed saponin compounds, *C. albicans* showed an increase in Pz and a decrease in phospholipase activity. This was also validated through the quantitative analysis of the genes encoding the phospholipase B protein. The above results indicate that tea seed saponin monomers inhibited the invasion of *C. albicans* by decreasing the expression of phospholipase B and suppressing the activity of secreted phospholipase B.

The abilities of morphological transformation and hyphal development are key to the interaction with the host and serve as the main factors contributing to the pathogenesis of *C. albicans*. These processes mainly occur during the invasion of *C. albicans* into the host. Figure 3C shows that 50 μΜ ASA and TE1 (1/2 MIC) inhibited the morphological transformation of *C. albicans*, while 100 μΜ TE2 also inhibited the growth of hyphae in yeast cells of *C. albicans*. This suggests that tea seed saponin monomers possess the potential to reduce the pathogenicity of *C. albicans* by inhibiting its morphological transformation. Furthermore, hyphal development plays an important role in biofilm formation, contributing to enhanced virulence through the evasion of host immunity [45]. Baillie et al. demonstrated that hyphae are a fundamental component for the complete development of mature biofilm structures [46]. Mature biofilms are more difficult to eradicate due to their complex structure. The extracellular matrix contains β-glucan synthase, which prevents the effect of antifungal drugs by binding with azole drugs [9]. Our research results indicate that ASA, TE1, and TE2 have both inhibitory effects on the formation of biofilms and destructive effects on mature biofilms. The relative expression levels of fungal-specific genes, such as *UME6*, *EED1*, *HGC1*, *HYR1*, *EFG1*, *ALS1*, etc., were downregulated by the three saponins at 100 μΜ.

In addition, the development of *C. albicans* hyphae is influenced by multiple signaling pathways. *RAS1* is a key modulator of hyphal growth, which transducts various signals to activate the cAMP–PKA and MAPK pathways. It is worth highlighting that these two pathways exhibit a synergistic relationship, further enhancing the morphological transformation process of *C. albicans* [47]. Li et al. showed that tea saponins suppressed *C. albicans* growth by lowering intracellular cAMP levels, as confirmed through cAMP supplementation tests and gene expression analysis of Ras1–cAMP–efg1 pathway-related genes [48]. In the current study, the relative expression levels of upstream core elements *RAS1*, *TPK2* in the cAMP–PKA pathways, and *CPH1* in the MAPK pathway were measured. It was found that ASA and TE1 significantly inhibited the expression of these genes during the adhesion stage. It is suggested that TE1 and ASA activated the catalytic subunit *TPK2* of PKA, which was involved in hyphae growth in the liquid culture medium, by modulating *RAS1* integration environmental conditions. Subsequently, phosphorylation activated *EFG1* in the cAMP–PKA pathway or *CPH1* in the MAPK signaling pathway, thereby inducing hyphal formation. In contrast, TE2 caused an augmentation in *RAS1* and *TPK2*, suggesting that TE2 might not induce hyphal development through *RAS1* to activate the cAMP–PKA pathway. Figure 6 shows that 100 μΜ ASA, TE1, and TE2 all exhibited a downregulating effect on the expression of *EFG1*. *EFG1* is considered the most important transcription factor involved in morphological transformation as it targets downstream and integrates signals from various upstream pathways to promote hyphal development [49]. *EFG1* is associated not only with the cAMP–PKA pathway but also with the N-acetylglucose pathway mediated by *NTG1* and the neutral pH pathway mediated by *RIM21*. The experimental conditions employed in this study to induce hyphal development consisted of a neutral pH environment with low nitrogen and a temperature of 37 °C. The low nitrogen and 37 °C incubation conditions induced hyphal development through the *RAS1*-dependent activation of the cAMP–PKA pathway, while TE2 showed *RAS1* independence. In summary, it is postulated that TE2 potentially modulates the transcriptional activity of *EFG1* via a separate signaling pathway, in contrast to TE1 and ASA perception. The signaling cascade is hypothesized to be triggered by the detection of a neutral pH environment. The exact underlying mechanism necessitates additional investigations (Figure 8).

In the investigation of the inhibitory impact of individual tea seed saponin monomers on *C. albicans*, significant differences in activity were observed among various monomers, both in the planktonic state and in the pathogenic hyphal form. Specifically, TE1 and TE2 were identified as isomers, differing in their substituent groups at the C22 and C28 positions. TE1 exhibited fungicidal activity against planktonic *C. albicans*, whereas TE2 did not demonstrate fungicidal properties within the range of experimental concentrations. Furthermore, notable disparities existed in their ability to impede the signaling pathway of hyphal development. Previous research indicated that TE1 and TE2 also manifested contrasting effects on individuals’ perceptions of sucrose sweetness [50]. The presence of an acetyl group at the C28 position augmented gastric protective activity, whereas the Ang group at the C22 position influenced saponin activity [51]. These findings serve as a foundation for asserting the superiority of ASA activity over TE1 and TE2 activity. Given this, it appears that the molecular structures of saponins are intricately linked to their biological activity. Further research is needed to study the structure–activity relationships of saponins, including their optimal substituent group, number of groups, and substitution position.

Tea seed saponins exhibit superior surfactant properties, including emulsification and foaming capabilities, making them a valuable addition to washing products in the chemical industry. Ren has successfully utilized tea saponins as non-ionic surfactants to substitute artificially synthesized surfactants in various products, such as toothpaste, water-free hand sanitizers, and fruit and vegetable cleaning agents. This substitution has yielded positive outcomes in terms of sterilization, stain removal, and oil elimination [52]. *C. albicans* demonstrates the ability to adhere to and create biofilms on both biological surfaces, such as the oral cavity and skin, and non-biological surfaces, including instruments and equipment. Given the promising anti-biofilm properties of tea seed saponin, coupled with its minimal toxicity and oral non-toxicity [53,54,55], there is potential for its utilization in human body care products and environmental disinfection and sterilization products.

## 4. Materials and Methods

### 4.1. Chemicals, Strains, and Growth Conditions

The *C. albicans* strain ATCC 10231 was cultured as described in the literature [14]. *C. albicans* strain ATCC 10231 was purchased from the Guangdong Provincial Microbial Culture Collection Center (Guangzhou, China) and cultured on yeast extract–peptone–dextrose agar (YPD, Sangon Biotech (Shanghai) Co., Ltd., Shanghai, China). Before each test, fresh cell suspensions were prepared by inoculating single colonies of the strain in YPD liquid media and incubated overnight at 30 °C at 200 rpm. Tea seed saponin monomers TE1, TE2, and ASA were created in the laboratory. RPMI 1640 medium and D-PBS were purchased from Hangzhou Gino Biomedical Technology Co., Ltd. (Hangzhou, China). XTT (2,3-bis (2-methoxy-4-nitro-5-sulfophenyl)-5-[(phenylamino)carbonyl]-2H-tetrazolium hydroxide) and sodium salt were obtained from Beijing Coolaber Technology Co., Ltd. (Beijing, China). The 0.5% crystal violet aqueous solution was purchased from Beijing Solaibao Technology Co., Ltd., and the 50% yolk lotion was purchased from Qingdao Haibo Biotechnology Co., Ltd. (Qingdao, China). Octane and menaquinone were purchased from Shanghai Macklin Biochemical Co., Ltd. (Shanghai, China).

### 4.2. Establishment of a Biofilm Model

The biofilm model was based on the recommended inoculation density in the literature [19]. *C. albicans* ATCC 10231 was cultured in YPD media until they reached the mid-log phase, followed by washing with sterile D-PBS and then dilution to a concentration of 2 × 10^6^ CFU/mL in RPMI 1640 medium. Subsequently, 200 µL of the fungal solution was added to each well of a 96-well plate and incubated at 37 °C for 1.5, 3, 6, 12, 24, 48, and 72 h. The plates were then rinsed twice with D-PBS and fixed with an XTT–menaquinone solution in the dark at 37 °C for 1 h. The absorbance value of the supernatant at OD490 was measured. Six independent biological replicates were performed for each time point.

A previous study provided a description of the preparation method for the XTT–menaquinone solution [56]. Specifically, the 0.5 mg/mL XTT solution was dissolved with D-PBS and mixed with a 10 mM menaquinone solution dissolved with acetone, resulting in a final concentration of menadione of 1 μM.

### 4.3. The Effects of ASA, TE1, and TE2 on the Adhesion Ability of C. albicans ATCC 10231

The determination of adhesion ability was conducted using a 96-well plate. RPMI 1640 medium containing fresh *C. albicans* cells (2 × 10^6^ CFU/mL) was mixed with different concentrations of ASA, TE1, and TE2 to achieve final concentrations of 0, 12.5, 25, 50, 100, 200, and 400 µM. The co-culture system, consisting of 200 µL of the mixture, was incubated at 37 °C for 1.5 h. Following incubation, the system was washed twice with D-PBS to remove non-adherent fungal cells. Subsequently, the system was stained with XTT solution for 1 h at 37 °C in the dark, and the resulting suspensions were measured using a Synergy H1 microplate reader (BioTek Instruments, Inc., Winooski, VT, USA) at OD490. The adhesion rate of *C. albicans* cells following drug treatment was calculated using the following formula:Adhesion% = (OD490_treatment group_ − OD490_blank_)/(OD490_negative control group_ − OD490_blank_) × 100%

Biofilm quantification was conducted using the CV staining assay with slight modifications [57]. Specifically, *C. albicans* cells were incubated with the aforementioned treatment for 1.5 h, followed by two washes with D-PBS to remove non-adherent cells. The wells were then air-dried for 45 min and fixed with 100 µL of a 0.5% aqueous solution of CV for 15 min. To fix the biofilms, D-PBS was used to remove excess CV through washing, followed by the addition of 100 µL of anhydrous ethanol to release the dyes from the biofilms. After 30 min, the absorbance value at 570 nm was measured to determine the biofilm inhibition effect. The above experimental results are expressed as the average of six replicates.

In addition, to investigate the impact of ASA, TE1, and TE2 on the morphological transformation of *C. albicans*, the *C. albicans* were incubated for 1.5 h according to the above treatment. Each well was washed twice with D-PBS to remove non-adherent cells, and the cellular morphology of *C. albicans* was observed using an inverted microscope. This experiment was repeated three times.

### 4.4. The Effects of ASA, TE1, and TE2 on Biofilm of C. albicans ATCC 10231

The biofilm formation experiment was performed on a 96-well plate following established protocols with minor adaptations [58]. RPMI 1640 medium containing fresh *C. albicans* cells (2 × 10^6^ CFU/mL) was added in 100 µL volumes to each well of the 96-well plate. The plate was then incubated at 37 °C for 1.5 h. Afterward, non-adherent cells were removed through washing with D-PBS, and fresh RPMI 1640 medium containing different concentrations of ASA, TE1, and TE2 was added. The final concentrations of ASA, TE1, and TE2 were 0, 12.5, 25, 50, 100, 200, and 400 µM. The plate continued to be incubated at 37 °C for 24 h.

In addition, a biofilm grown for 48 h in a 96-well plate was used to detect the impact on mature biofilms. RPMI 1640 medium containing fresh *C. albicans* cells (2 × 10^6^ CFU/mL) was added in 100 µL volumes to each well of the 96-well plate. The plate was then incubated at 37 °C for 24 h. After incubation, the plate was washed twice with D-PBS to remove suspended cells, and fresh RPMI 1640 medium containing different concentrations of ASA, TE1, and TE2 was added. The final concentrations of ASA, TE1, and TE2 were 0, 25, 50, 100, 200, 400, and 800 µM. The plate was then incubated at 37 °C for an additional 24 h. 

The metabolic activity of the biofilm was measured using the XTT reduction assay, while the quantification of the biofilm was conducted using the CV staining assay. The experimental results are represented as the average of six replicates.

### 4.5. Determination of Cell Surface Hydrophobicity (CSH)

The determination of CSH is described in reference [59]. RPMI 1640 medium containing fresh *C. albicans* cells (1 × 10^7^ CFU/mL) was inoculated into a 6-well flat plate, and different concentrations of ASA, TE1, and TE2 were added at 0 and 24 h, resulting in final concentrations of 0, 25, 50, and 100 µM for ASA, TE1, and TE2, respectively. Co-culturing with ASA, TE1, and TE2 at 0 h for 1.5 h was conducted to investigate the effect of tea seed saponin monomers on the CSH of adherent cells. Additionally, co-culturing with ASA, TE1, and TE2 for 24 h was performed to investigate the effect of tea seed saponin monomers on the CSH of mature biofilm cells. Subsequently, *C. albicans* cells were collected from each treatment group at 3000× *g* and centrifuged for 5 min. The cultured cells were then rinsed with D-PBS three times and diluted to OD600 of 1.0. To create the mixture, 1.2 mL of cell suspension and 0.3 mL of n-octane were combined in a clean, covered glass test tube. The mixture was subjected to vortexing for a duration of 3 min, followed by a standing period of 30 min. The OD600 of the aqueous phase was measured using a Synergy H1 microplate reader. The CSH% was determined using the following formula:CSH% = (OD600_control group_ − OD600_treatment group_)/OD600_control group_ × 100%
where the OD600 control group represents the OD600 without the addition of n-octane. This experiment was repeated three times.

### 4.6. Extracellular Phospholipase Assay

According to the report by Yousuf et al. [60], phospholipase levels were measured in both adhesion-stage and mature biofilm-stage cells using egg yolk emulsion agar. In the adhesion stage, RPMI 1640 medium containing fresh *C. albicans* cells (1 × 10^5^ CFU/mL) was used, and different concentrations of ASA, TE1, and TE2 were added to the fungal solution. In the mature biofilm stage, RPMI 1640 medium containing fresh *C. albicans* cells (1 × 10^5^ CFU/mL) was used. The cells were incubated at 37 °C for 24 h, and then different concentrations of ASA, TE1, and TE2 were added to the mature biofilm. In each treatment group, 1 µL of *C. albicans* cell suspension was evenly distributed onto egg yolk emulsion agar and incubated at 37 °C for 4 days. The presence of sedimentation circles around the colonies indicated the production of phospholipase by the *C. albicans* strain. The colony diameter (d1) and the diameter of the precipitation zone (d2) were measured, and the ratio of d1/d2 represented the phospholipase activity, expressed as the Pz value. A negative correlation was observed between the Pz value and phospholipase activity, with lower Pz values indicating higher phospholipase activity. A Pz value of 1 signifies the absence of phospholipase activity in the strain, while a Pz value ranging from 0.90 to 0.99 indicates an extremely low level of phospholipase activity. Similarly, a Pz value ranging from 0.80 to 0.89 suggests a low level of phospholipase activity, whereas a Pz value ranging from 0.70 to 0.79 indicates a high level of phospholipase activity. Finally, a Pz value of less than or equal to 0.69 signifies an extremely high level of phospholipase activity. The phospholipase experiment was conducted independently on three occasions.

### 4.7. Real-Time Quantitative Reverse Transcription PCR (qRT-PCR) of Virulence Factor Genes Related to C. albicans ATCC 10231

Sample preparation involved adjusting fresh *C. albicans* cells in the logarithmic growth phase to a concentration of 1 × 10^7^ CFU/mL using RPMI 1640 medium. ASA, TE1, and TE2 were added at a final concentration of 100 µM, while the control group received no drugs. The fungal cells were then cultivated at 37 °C and 200 rpm for 1.5 h, after which they were collected through centrifugation. Each treatment group consists of three independent biological replicates.

The yeast RNA extraction kit (Aidlab Biotech, Beijing, China) was used to extract total RNA from the above treatment groups and control groups. The concentration and purity of RNA were verified by agarose gel electrophoresis and UV spectrophotometry. Using SYBR^®^ Premix Ex Taq™ (Monad Biotech, Shanghai, China), total RNA was reversed transcribed into cDNA.

Using *ACT1* as the reference gene, 16 genes related to *C. albicans* virulence factors were detected via qRT-PCR using paired primers (Table 1). The PCR program was as follows: 95 °C for 10 min, followed by 40 cycles of 95 °C for 10 s, 60 °C for 10 s, and 72 °C for 15 s. The relative expression levels of the genes were normalized using the 2^−ΔΔCT^ method.

### 4.8. cAMP Rescue Test

To test the effect of cAMP on hyphal formation after ASA, TE1, and TE2 treatment, RPMI 1640 medium containing fresh *C. albicans* cells (2 × 10^6^ CFU/mL) was mixed with different concentrations of ASA, TE1, and TE2 to achieve final concentrations of 0, 25, 50, and 100 µM. Then, db-cAMP was added to the drug treatment group and the drug-free treatment group at a final concentration of 1 mM. The db-cAMP-free group served as the control. The co-culture system, consisting of 200 µL of the mixture, was incubated at 37 °C for 4 h. Following incubation, the system was washed twice with D-PBS to remove non-adherent fungal cells, and the cellular morphology of *C. albicans* was observed using an inverted microscope. This experiment was repeated three times.

### 4.9. Statistical Analysis

All data are presented as the mean ± standard deviation. The results were analyzed with SPSS Version 22.0 using one-way ANOVA to demonstrate the significant differences (*p* < 0.05, *p* < 0.01). GraphPad Prism (v9.4.1) was used to process the data and generate the figures.

## 5. Conclusions

This study reveals that the tea seed saponin monomers ASA, TE1, and TE2 can impede the development of *C. albicans* biofilms and impair fully formed biofilms, and the order of inhibitory effectiveness was found to be ASA > TE1 > TE2. The mechanism underlying this inhibitory effect is associated with the attenuation of crucial virulence factors of *C. albicans*, including cell surface hydrophobicity, adhesion ability, hyphal morphology transformation, and phospholipase activity. Furthermore, the inhibition of *C. albicans* growth by ASA and TE1 was attributed to the suppression of *RAS1* activation, consequently impeding the cAMP–PKA and MAPK signaling pathways. Conversely, TE2 is likely to modulate the morphological transformation and hyphal growth of *C. albicans* via alternative pathways that do not rely on *RAS1*. This study mainly focused on the single biofilm composed of *C. albicans* ATCC 10231; the effect of tea seed saponins on the complex biofilm of *C. albicans* remains to be studied in more depth in the future.

## Figures and Tables

**Figure 1 ijms-25-03599-f001:**
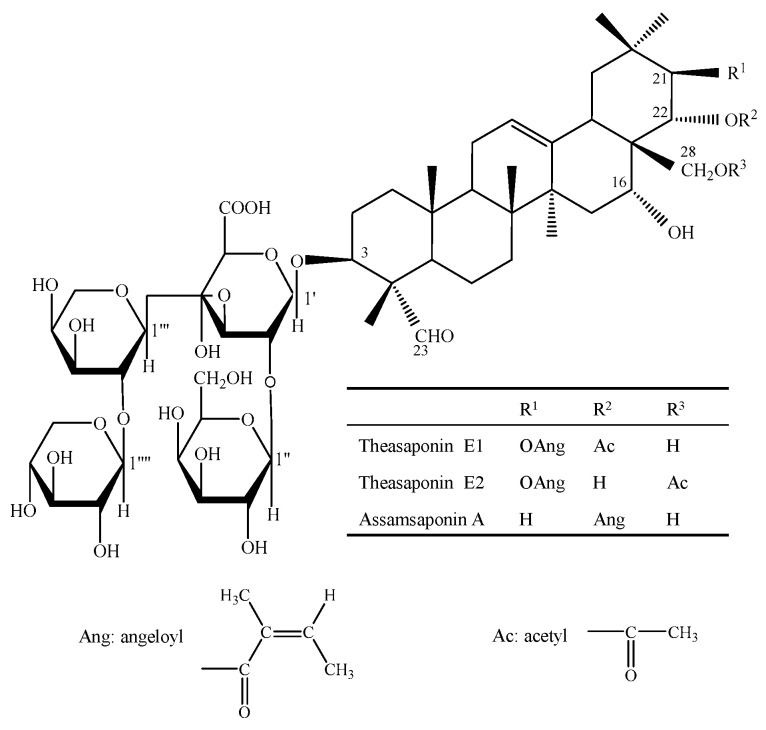
The structures of assamsaponin A (ASA), theasaponin E1 (TE1), and theasaponin E2 (TE2).

**Figure 2 ijms-25-03599-f002:**
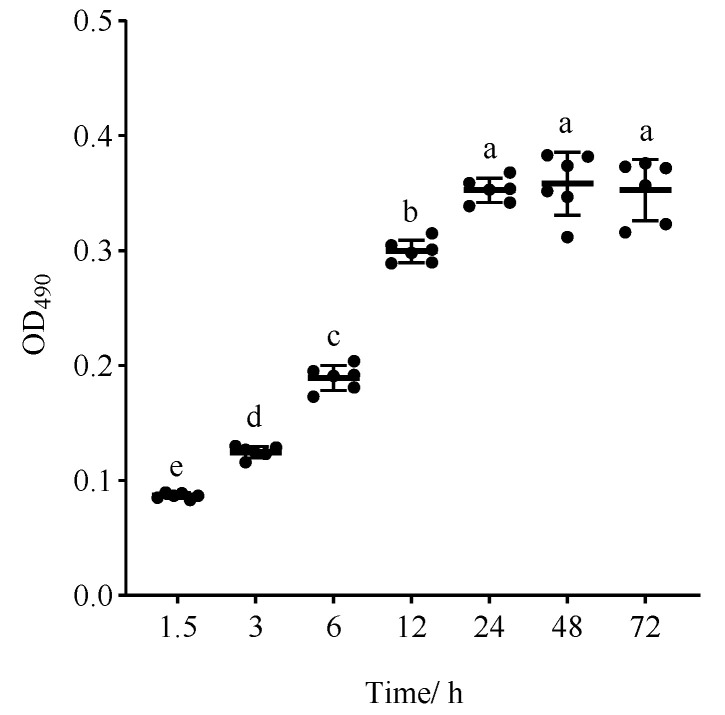
The biofilm growth curve for the biofilm formation of *C. albicans* ATCC 10231. Different letters indicate significant differences (*p* < 0.05).

**Figure 3 ijms-25-03599-f003:**
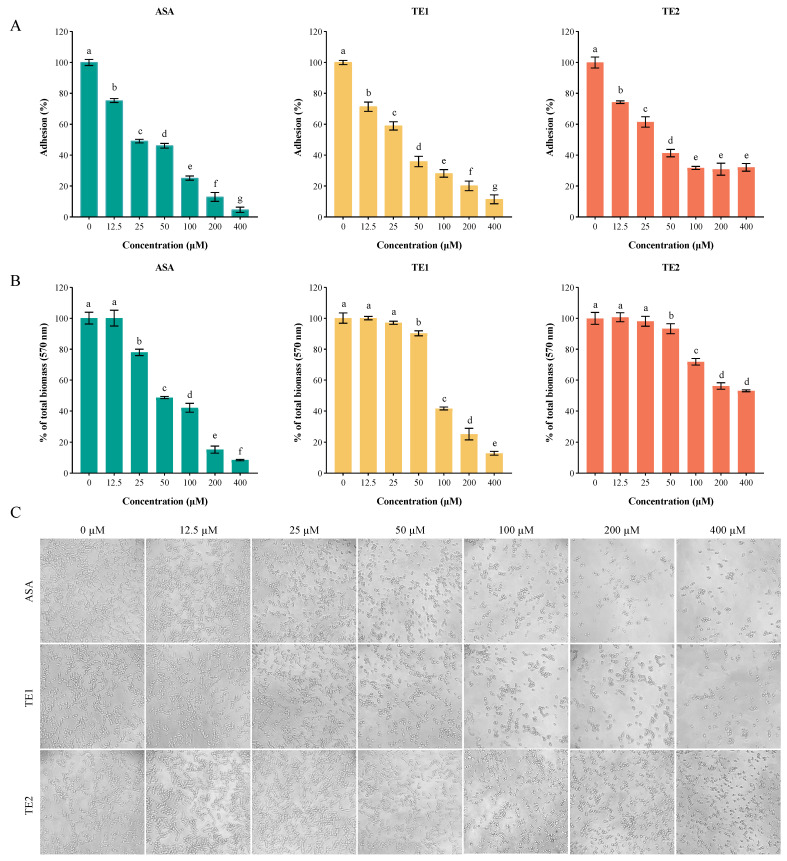
ASA, TE1, and TE2 inhibited the adhesion and biofilm growth of *C. albicans* ATCC 10231. (**A**) Quantitative analysis of the effect of different concentrations of ASA, TE1, and TE2 on the adhesion of *C. albicans* using the XTT reduction assay; (**B**) quantitative analysis of the biofilm growth of *C. albicans* by different concentrations of ASA, TE1, and TE2 during the adhesion stage using the crystal violet staining (CV) assay; (**C**) microscopic representations of the effect of different concentrations of ASA, TE1, and TE2 on the adhesion of *C. albicans* (100X). The data represent the average (±standard deviation, SD) of six independent experiments. Different letters above the column indicate significant differences (*p* < 0.05).

**Figure 4 ijms-25-03599-f004:**
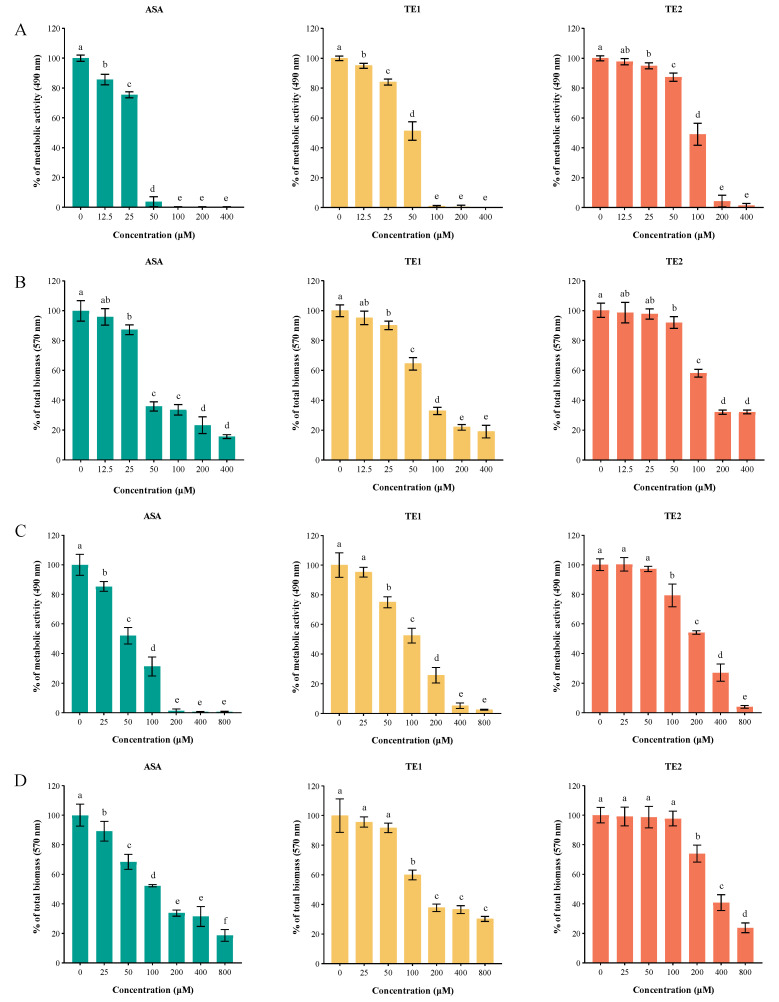
Inhibition of biofilm formation and eradication of mature biofilm by ASA, TE1, and TE2 against *C. albicans* ATCC 10231. (**A**,**C**) Quantitative detection of different concentrations of ASA, TE1, and TE2 on the cellular metabolic activity of early biofilm formation and mature biofilm formation of *C. albicans* via the XTT reduction assay; (**B**,**D**) quantitative detection of different concentrations of ASA, TE1, and TE2 on the total biomass of early biofilm formation and mature biofilm of *C. albicans* via the CV staining assay. The data represent the average (±standard deviation, SD) of six independent experiments. Different letters above the columns indicate significant differences (*p* < 0.05).

**Figure 5 ijms-25-03599-f005:**
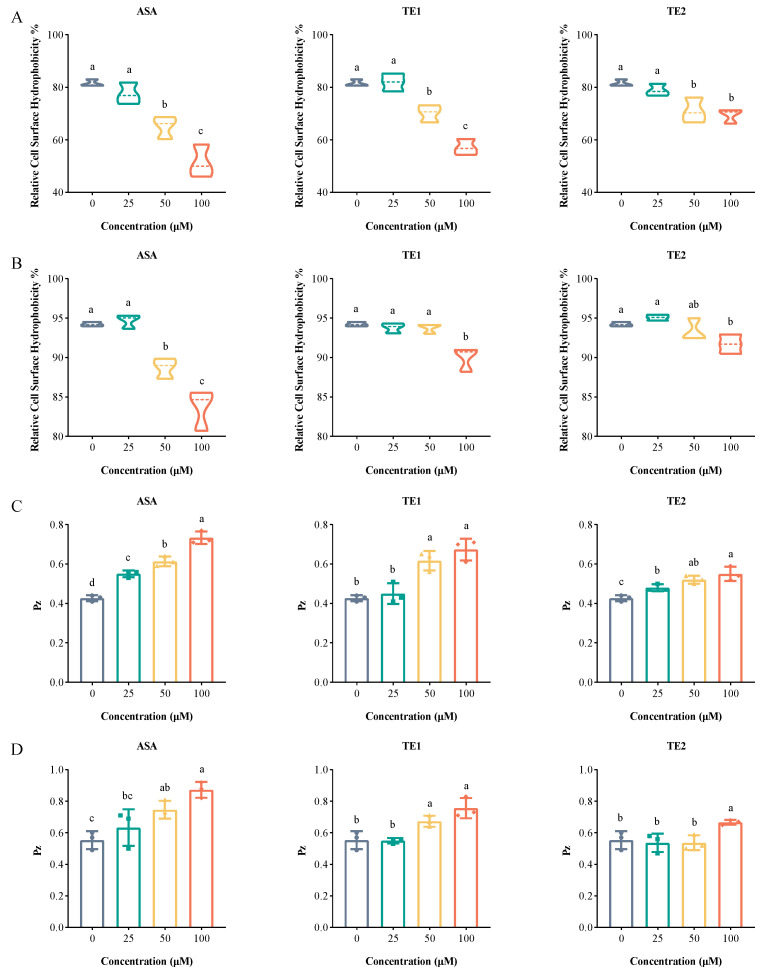
Effects of ASA, TE1, and TE2 on the cell surface hydrophobicity and phospholipase production of *C. albicans* ATCC 10231. (**A**,**B**) The effect of different concentrations of ASA, TE1, and TE2 on the cell surface hydrophobicity of adhesive cells and mature biofilm cells; (**C**,**D**) the effect of different concentrations of ASA, TE1, and TE2 on the phospholipase production of adhesive cells and mature biofilm cells. The data represent the average (±standard deviation, SD) of three independent experiments, with each experiment repeating three measurements. The different letters above the bar charts indicate significant differences (*p* < 0.05).

**Figure 6 ijms-25-03599-f006:**
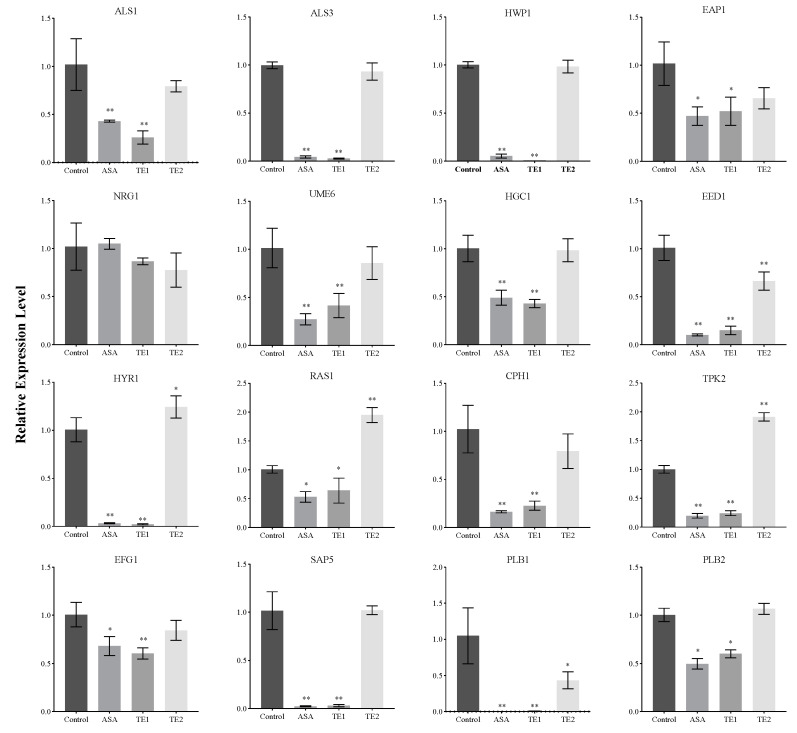
Effect of ASA, TE1, and TE2 on the expression of virulence factor-related genes in *C. albicans* ATCC 10231. The data represent the average (±standard deviation, SD) of three independent experiments. Compared with the controls, * *p* < 0.05 indicates a significant difference, and ** *p* < 0.01 indicates a highly significant difference.

**Figure 7 ijms-25-03599-f007:**
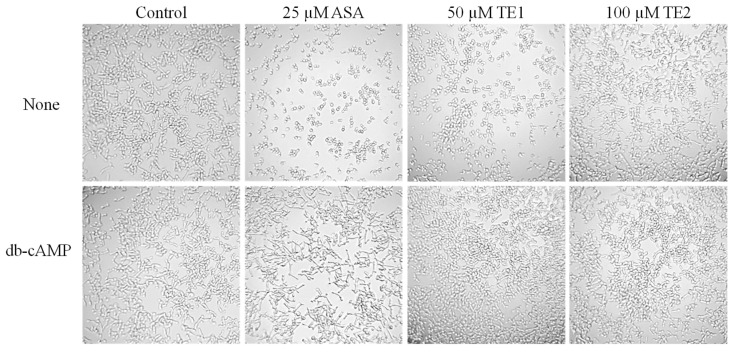
The effect of cAMP on hyphae with ASA, TE1, and TE2 treatment on *C. albicans* ATCC 10231 (20X).

**Figure 8 ijms-25-03599-f008:**
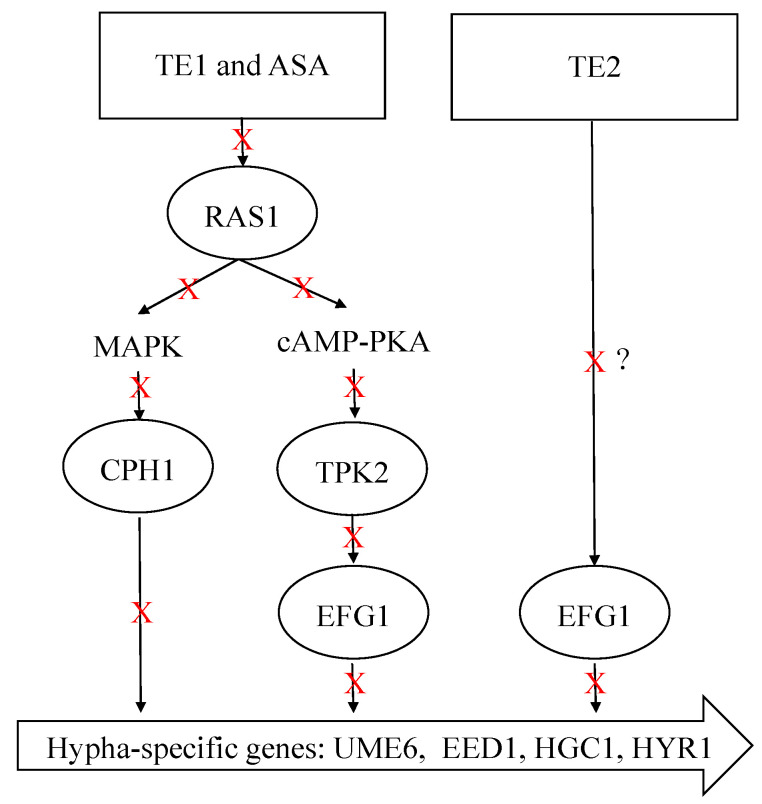
Potential mechanisms of inhibition of hyphal formation by ASA, TE1, and TE2. The red Xs and black arrows represent downregulation or inhibition. The symbol “?” represents that the mechanism is not yet clear.

**Table 1 ijms-25-03599-t001:** Primers for qRT-PCR.

Primer	Sequence
ACT1-F	TGACCGAAGCTCCAATGAATCC
ACT1-R	CCGGTGGTTCTACCAGAAGAGT
ALS1-F	CCTATGCCACCACTACCACTGT
ALS1-R	AATCGGAGGTTGTGCTGTTGAC
ALS3-F	CGCAACCACCACTACCATTACC
ALS3-R	CACCTGGAGGAGCAGTGATTGT
HWP1-F	ACAGGTAGACGGTCAAGGTGAA
HWP1-R	TGAGGTGGATTGTCGCAAGGT
EAP1-F	TGTGATGGCGGTTCTTGTTCTC
EAP1-R	GTGGACTCGGTAGCTGGTGTAG
NRG1-F	TGCTAGTGCTGCTGGTAGTACA
NRG1-R	CTGCTGCTGCTTGGTTGGTATT
UME6-F	TGGTGGTGTCAGTGTTAGTGCT
UME6-R	TTGGTGGTGGTGGAAGAGAAGG
HGC1-F	GCAACCACCACCACCAATGAA
HGC1-R	ACAGCACGAGAACCAGCGATA
EED1-F	TGCTCTACCACCACAACAAG
EED1-R	TTGCGGTGCTTGCTCATA
HYR1-F	GCTCAGGCTCAGGCTCACAA
HYR1-R	TTCGGAACCAGAACCAGAACCA
TPK2-F	TTCAACAACCGCAGCAACAACT
TPK2-R	TTCAGCAGCCGATTTGGAAACA
RAS1-F	TGGTGGTGTAAGTAGTGATGGA
RAS1-R	GTTGTTGCTGTTGTTGTTGTCT
EFG1-F	TTCACAACAGCCACCACTACCA
EFG1-R	TGCTCTTCTGACAACCGACACA
CPH1-F	TGCTGCCACTGCTCCAATGTA
CPH1-R	TGCTGCTGCTGCTGTTGTTG
SAP5-F	CCGTCGATGAGACTGGTAGAGA
SAP5-R	GGGAAGTGCGGGAAGATGCT
PLB1-F	TGGTCGTCCAACTGGTAAGTGT
PLB1-R	GCCATCTTCTCCACCGTCAAC
PLB2-F	ACATGCCAATCCCACCATTCCT
PLB2-R	TGACCGTCTTCACCTCCATCTG

## Data Availability

The authors declare that all data generated or analyzed during this study are included in this published article. Related data are available from the authors upon reasonable request.

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
