# Peer review of "Anti-Biofilm Activity of Assamsaponin A, Theasaponin E1, and Theasaponin E2 against Candida albicans"

_ijms, 2024, doi:10.3390/ijms25073599_

Round 1

Reviewer 1 Report

Comments and Suggestions for Authors

The manuscript presents a comprehensive investigation into the anti-biofilm activities of tea seed saponin monomers—assamsaponin A, theasaponin E1, and theasaponin E2—against Candida albicans. This study elucidates the potential of these compounds in inhibiting biofilm development, affecting the organism's metabolism, and modulating virulence gene expression. The findings underscore the promising role of tea seed saponins as novel agents in the battle against C. albicans biofilms. Nevertheless, to augment the manuscript's contribution to the field, the following points merit further attention:

1. The Introduction provides a necessary backdrop for the study but would benefit from a more streamlined synthesis of the literature. A focused discussion on current therapeutic strategies against C. albicans, their limitations, and the urgent need for novel antifungal agents could better contextualize the study's significance.

2. The rationale for employing an inoculum density of 2 × 10^6 CFU/mL is not explicitly stated. Clarification on how this specific concentration aligns with the study's objectives and its relevance to clinical or environmental scenarios would strengthen the methodological foundation.

3. The paper seems to emphasize localized C. albicans infections, which suggests a potential for topical therapeutic applications. It would be insightful to explore how topical application method could influence the efficacy to mimick real-world treatment scenarios.

4. While the study effectively examines biofilm inhibition, incorporating an analysis of biofilm architecture could provide deeper insights into how these saponins affect C. albicans morphology and structural integrity.

5. For the findings to have translational relevance, it is crucial to address the stability and bioavailability of tea seed saponins within biological systems.

Comments on the Quality of English Language

Language editing is needed for the paper.

Reviewer 2 Report

Comments and Suggestions for Authors

Overall, while the study provides valuable insights into the anti-biofilm activity of tea seed saponin monomers against C. albicans, addressing the aforementioned critical review points would strengthen the scientific rigor, interpretation, and translational potential of the findings.

Points for Improvement:

Ensure consistency in assay conditions such as incubation time, temperature, and media composition to minimize variability and facilitate accurate comparisons between experiments.

Quantify the results of the assays (e.g., XTT reduction assay, crystal violet staining assay) to provide objective measurements of biofilm formation and eradication. This will enhance the reliability and interpretability of the findings.

Validate the mechanistic insights into the anti-biofilm activity of tea seed saponin monomers through additional experimental approaches, such as protein expression analysis or genetic manipulation of relevant signaling pathways.

Identify the specific molecular targets of tea seed saponin monomers within Candida albicans cells using techniques such as proteomics or transcriptomics. This will elucidate the precise mode of action and enhance understanding of their therapeutic potential.

Assess the cytotoxicity of tea seed saponin monomers on host cells using appropriate cell viability assays. This is crucial for determining their safety profile and potential for clinical use.

Investigate potential synergistic or antagonistic interactions between tea seed saponin monomers and existing antifungal agents to enhance efficacy and minimize the risk of resistance development.

Round 2

Reviewer 1 Report

Comments and Suggestions for Authors

The manuscript can be accepted in the current form.